# 3D Hollow rGO Microsphere Decorated with ZnO Nanoparticles as Efficient Sulfur Host for High-Performance Li-S Battery

**DOI:** 10.3390/nano10091633

**Published:** 2020-08-20

**Authors:** Zhi Zhang, Zichuan Yi, Liming Liu, Jianjun Yang, Chongfu Zhang, Xinjian Pan, Feng Chi

**Affiliations:** School of Electronics and Information, University of Electronic Science and Technology of China, Zhongshan Institute, Zhongshan 528402, China; liulmxps@126.com (L.L.); sdyman@uestc.edu.cn (J.Y.); cfzhang@uestc.edu.cn (C.Z.); xinjpan@163.com (X.P.); chifeng@semi.ac.cn (F.C.)

**Keywords:** ZnO, graphene, spray drying, hollow structure, lithium-sulfur battery

## Abstract

Lithium-sulfur battery (LSB) will become the next generation energy storage device if its severe shuttle effect and sluggish redox kinetics can be effectively addressed. Here, a unique three-dimensional hollow reduced graphene oxide microsphere decorated with ZnO nanoparticles (3D-ZnO/rGO) is synthesized to decrease the dissolution of lithium polysulfide (LiPS) into the electrolyte. The chemical adsorption of ZnO on LiPS is combined with the physical adsorption of 3D-rGO microsphere to synergistically suppress the shuttle effect. The obtained 3D-ZnO/rGO can provide sufficient space for sulfur storage, and effectively alleviate the repeated volume changes of sulfur during the cycle. When the prepared S-3D-ZnO/rGO was used as the cathode in LSB, an initial discharge specific capacity of 1277 mAh g^−1^ was achieved at 0.1 C. After 100 cycles, 949 mAh g^−1^ can still be maintained. Even at 1 C, a reversible discharge specific capacity of 726 mAh g^−1^ was delivered.

## 1. Introduction

Lithium-ion battery (LIB) is the most widely used rechargeable battery, with many advantages: high energy density, long service life and low cost [1,2,3]. After development for more than 20 years, the specific capacity of the commercial cathode material is close to its theoretical value, which still cannot meet the growing energy need. Therefore, the search for a new rechargeable battery with high energy density has become urgent [4,5]. Recently, a lithium-sulfur battery (LSB) has caught the attention of researchers all over the world. Elemental sulfur has a high theoretical specific capacity of 1675 mAh g^−1^, indicating its great potential as energy storage material. The low working voltage of LSB (~2.2 V) can adapt to the commercial need. In addition, sulfur also has the advantage of being a sufficient resource, having low cost, and being harmless to the environment [6,7,8,9,10,11]. Therefore, LSB is regarded as the most potential substitute for commercial LIB, but the performance of LSB is still difficult to reach the current level of commercial LIB, for these reasons: (1) the low conductivity of elemental sulfur and its discharge products (Li_2_S_2_ and Li_2_S). (2) The dissolution of the reaction intermediate (Li_2_S_n_, 4 ≤ n ≤ 8) into the electrolyte. (3) The repeated volume changes of elemental sulfur during the cycling process [12,13,14].

In recent years, researchers have conducted a lot of work on the modification of cathode materials for the improved performance of LSB. The use of the carbon-sulfur composite cathode is considered to be an effective way to realize the improvement [15,16]. Among the carbon materials, graphene has received extensive attention. Graphene has many advantages, such as high specific surface area, excellent electrical conductivity, high structural stability, high strength, high flexibility, and easy functionalization [17,18]. Graphene as a sulfur host can improve the conductivity of the cathode. The good flexibility of graphene makes it possible to effectively relieve the volume change of the sulfur. For example, the porphyrin-derived graphene-based nanosheet was prepared by Zhang et al. [19] When used as a sulfur host, superb cycling stability was realized. However, graphene can only immobilize lithium polysulfide (LiPS) by weak physical adsorption, which cannot effectively suppress shuttle effect [20,21]. A recent study shows that polar metal oxides (Co_3_O_4_, Fe_3_O_4_, NiO, TiO_2_, ZnO et al.) can confine LiPS by O-S bond [22]. This strong chemical bond can significantly inhibit the dissolution of LiPS, thereby enhancing cycling stability. For example, Wang et al. reported that the N-Co_3_O_4_@N-C/rGO can produce an obvious effect for LiPS absorption [23]. The TCS/ZnO is prepared by Gui et al. and promoted limitation on LiPS was achieved [24]. However, the further research of metal oxide is hindered by the poor conductivity and the tendency to agglomerate during the preparation process [25,26,27,28,29,30,31]. Therefore, the metal oxide and the conductive carbon material can be combined to make a long complement.

Herein, a unique three-dimensional hollow reduced graphene oxide microsphere decorated with ZnO nanoparticles (3D-ZnO/rGO) is synthesized by the sol-gel method, following spray drying and calcination. After sulfur loading, the obtained S-3D-ZnO/rGO is used as the cathode. The introduced GO can produce numerous attachment sites for the formation of ZnO, thereby effectively inhibiting the agglomeration of ZnO. After spray drying, the ZnO/GO sheets are textured to form a three-dimensional hollow structure. The obtained 3D-ZnO/GO can provide sufficient space for sulfur storage and effectively alleviate the volume change of sulfur. To further improve the conductivity of 3D-ZnO/GO, calcination is conducted to remove the oxygen-containing functional group. When the synthesized 3D-ZnO/rGO is used as a sulfur host, excellent cycling stability and outstanding rate performance are achieved.

## 2. Materials and Methods

### 2.1. Synthesis

First, 21.2 g of GO dispersion was uniformly dispersed in 30 mL of methanol by ultrasonication. Subsequently, 0.2318 g of Zn(CH_3_COO)_2_·2H_2_O was put into 50 mL of methanol. The two solutions were mixed by stirring. The obtained solution was sealed and heated to 60 °C. Then, 0.1426 g of KOH was put into 40 mL of methanol at 60 °C for obtaining a KOH/methanol solution. Subsequently, the KOH/methanol solution was dropwise added to the above-mentioned solution. The resulting solution was continuously stirred at 60 °C for 2 h. The resulting precipitate was centrifuged to obtain the ZnO/GO composite. The synthesized ZnO/GO composite was dispersed in deionized water to obtain a ZnO/GO dispersion with a concentration of 3 mg mL^−1^ as spray drying precursor. The precursor was spray dried at 200 °C for 3D-ZnO/GO composite. The prepared 3D-ZnO/GO composite was heated to 800 °C under Ar for 2 h, to realize the preparation of 3D-ZnO/rGO. As a comparison, ZnO/rGO was synthesized by the same method, except the process of spray drying. The obtained 3D-ZnO/rGO and sulfur powder were uniformly ground with a mass ratio of 3:1. The mixture was then sealed in a Teflon-lined stainless steel reaction kettle and heated to 155 °C for 12 h, resulting in the S-3D-ZnO/rGO composite.

### 2.2. Sample Characterization

The phase of product was explored by X-ray diffraction (XRD, D8 Discover, Bruker, Karlsruhe, Germany). Raman (inVia Reflex) was used to evaluate sample composition and structural defects in its carbonaceous materials. The morphologies of the synthesized product were characterized by scanning electron microscopy (SEM, Hitachi S-4800, Tokyo, Japan) and transmission electron microscopy (TEM, JEOL, JEM-2100F, Tokyo, Japan). The thermogravimetric analysis was conducted by SDT-Q-600. X-ray photoelectron spectroscopy (XPS, ESCALAB 250XI, Thermo Fisher Scientific, Waltham, MA, USA) was used to characterize the element content and bonding of the sample surface.

### 2.3. Electrochemical Measurements

The S-3D-ZnO/rGO, super P (Guotai Huarong Chemical New Material Co., Ltd. China) and polyvinylidene fluoride (PVDF, Aladdin Reagent Co., Ltd. Shanghai, China) were mixed according to a mass ratio of 8:1:1. N-methylpyrrolidone (NMP, Aladdin Reagent Co., Ltd. Shanghai, China) was used to form a uniform slurry with a certain viscosity. The slurry was coated on the carbon-containing aluminum foil by a doctor blade. After being dried, the aluminum foil was cut into disks with diameters of 10 mm, and the disks were pressed at a pressure of 10 MPa for 1 min. The weight of the active material content on the disks was weighed using a balance. The sulfur content per unit area is between 1.27 and 1.88 mg cm^−2^. The prepared disk was used as the cathode, the lithium foil (Qinhuangdao Lithium Co., Ltd., Qinhuangdao, China) was used as the counter electrode, the Celgard 2400 film (Shenzhen Weifeng Electronics Co., Ltd., Shenzhen, China) was used as a separator. Moreover, 1 M LiTFSI in 1,2-dimethoxy ethane (DME) and 1,3-dioxolane (DOL) (1:1 v/v) with 0.1 M LiNO_3_ were used as the electrolyte (Guotai Huarong Chemical New Material Co., Ltd., Zhangjiagang, China). The assembly of the button cell (CR2032) was conducted in the Ar-filled glove box. The Neware battery tester (BTS 4000, Neware Inc., Shenzhen, China) was used to evaluate the cycling stability and rate performance. The electrochemical impedance spectroscopy (EIS) and cyclic voltammetry (CV) were explored by electrochemical workstation (Princeton Applied Research, Versa STAT3, Ametek, PA, USA).

### 2.4. Theoretical Calculations

The Vienna ab initio simulation package (VASP) was employed to conduct the DFT calculation, and the results were visualized in Materials Studio. A plane-wave cutoff of 400 eV was set. The Li_2_S_6_ adhesion bonding energy (Eb) was calculated as follows: E_b_ = E_Li2S6/ZnO_ – E_Li2S6_ – E_ZnO_, where E_Li2S6/ZnO_, E_Li2S6_, and E_ZnO_ represent the total energy of the system, isolated Li_2_S_6_ and ZnO, respectively.

## 3. Results

The ZnO/GO composite was prepared by the sol-gel method at first (Figure 1). Subsequently, the ZnO/GO sheets were bent and stacked under the action of spray drying to form a hollow sphere structure, which is named 3D-ZnO/GO. In order to further improve the conductivity of ZnO/GO, high temperature reduction is carried out to remove a large number of oxygen-containing groups on GO, resulting in 3D-ZnO/rGO. Finally, 3D-ZnO/rGO is composited with S by simple melt diffusion method to synthesize S-3D-ZnO/rGO.

For comparison, pure ZnO is prepared by the sol-gel method. As shown in Figure 2a, the agglomeration in pure ZnO is serious, resulting in low specific surface area and low adsorption capacity for sulfur and LiPS. The SEM image of the flaky ZnO/rGO is performed in Figure 2b. The graphene sheets are stacked on top of each other, and a large amount of ZnO particles are evenly distributed on graphene. The graphene can provide a large number of active sites for heterogeneous nucleation, and the growth of ZnO nanoparticles and the uniform dispersion of ZnO nanoparticles are achieved. At the same time, graphene can enhance the electron conductivity, and its good flexibility can improve the structural stability of the cathode. However, the two-dimensional structure formed by the simple stack reduces the effective reaction area, resulting in a decrease in the effective utilization of ZnO. The SEM images of 3D-ZnO/rGO are performed in Figure 2c,d, the originally pleated graphene is crimped into spheres by spray drying, and their diameters are distributed in the range of 2–5 μm. The surface of the microsphere is evenly distributed with a large amount of ZnO nanoparticles. It should be pointed out that the hollow structure can achieve the high sulfur loading and reduce the volume change of the sulfur through the internal space of the hollow spherical structure. The EDS mappings in Appendix A further confirm the uniform distribution of carbon, oxygen, and zinc in 3D-ZnO/rGO and ZnO/rGO.

The TEM image of the 3D-ZnO/rGO is displayed in Figure 3a. The hollow structure is obviously observed, and ZnO nanoparticles are densely distributed on the surface of microsphere. The TEM image obtained by magnifying the edge of the microsphere is shown in Figure 3b. ZnO nanoparticles are explored more clearly with the size of 20–60 nm. The small amount of large diameter ZnO particles may be due to the fact that the graphene dispersion contains some water and the ZnO is hydrophilic, so the ZnO particles synthesized during the reaction tend to agglomerate to form large ZnO particles. It can be seen from Figure 3c and Appendix A that the ZnO particles are composed of many fine ZnO crystal grains, wherein the lattice spacing of 0.281 and 0.249 nm corresponds to the (100) and (002) crystal faces, respectively. The SAED pattern of ZnO shown in Figure 3d shows a plurality of diffraction rings, indicating the polycrystalline properties of the ZnO particles.

The SEM results of S-ZnO/rGO are shown in Figure 4a,b, the three-dimensional fold structure formed by the interweaving of graphene sheets was well preserved, and no obvious sulfur blocks were observed, indicating the uniform sulfur loading, which is also performed in Figure 4c. Similarly, after sulfur loading, spherical morphology and distinct graphene sheet fold structure are still maintained (Figure 4d,e). Compared with the 3D-ZnO/rGO microspheres, ZnO nanoparticles are difficult to observe, indicating that the elemental sulfur in the molten state diffuses evenly during the hydrothermal process. The element distribution of S is characterized on the microsphere, which can prove that S is uniformly distributed (Figure 4f). Furthermore, element distributions of Zn and C prove that the ZnO particles remain uniform on the graphene surface after hydrothermal reaction.

The XRD spectra are performed in Figure 5a. It can be seen that the XRD patterns have weak characteristic peaks of carbon in ~23.5° and ~43.0°, corresponding to the (002) and (100) crystal faces of crystalline graphite, respectively. The observed peaks of crystalline graphite prove that the introduced graphene oxide is successfully reduced. Furthermore, diffraction peaks at about 32.0°, 34.6°, 36.4°, 47.7°, 56.8°, 63.0°, 66.3°, 68.1°, and 69.2° match the characteristic peaks of ZnO (JCPDS, No. 36-1451), indicating that ZnO is successfully prepared. No other impurity peaks are observed, which indicates that the synthesized product is of high purity. After sulfur loading, the XRD patterns show significant changes (Figure 5b). The strong diffraction peaks of sulfur (JCPDS, No. 08-0247) are performed, proving the successful introduction of sulfur; no other obvious side effects occur.

Raman spectra are shown in Figure 6a. It can be seen that all three samples have a weak peak at about 430 cm^−1^, which corresponds to the ZnO crystal. Two distinct broad peaks at around 1344 and 1580 cm^−1^ are attributed to the D peak and G peak of graphene. The calculation results show that the *I*_D_/*I*_G_ values of ZnO/rGO and 3D-ZnO/rGO composites are both 1.03, which is significantly higher than the *I*_D_/*I*_G_ value (0.95) of ZnO/GO composites. The result shows that GO can be effectively reduced by high-temperature. After sulfur loading, two additional peaks at 217 and 472 cm^−1^ are detected in S-ZnO/rGO and S-3D-ZnO/rGO (Figure 6b), which is due to the S-S bond vibration of elemental sulfur [32]. At the same time, the calculation shows that the *I*_D_/*I*_G_ values of S-ZnO/rGO and S-3D-ZnO/rGO composites are 1.08 and 1.10, respectively, which shows a little increase. The increase shows that the degree of disorder of the obtained rGO is promoted, which may be related to the bonding between graphene and sulfur.

In order to measure the sulfur loading in S-ZnO/rGO and S-3D-ZnO/rGO composites, TGA test was conducted in N_2_ from room temperature to 600 °C (Figure 7). The weight loss occurs at about 200 °C. With the temperature increases, the elemental sulfur in the composite gradually sublimates. The decrease ends at about 310 °C, and the residual product is pure ZnO/rGO. After calculation, the sulfur contents in S-ZnO/rGO and S-3D-ZnO/rGO are 66.4 wt.% and 69.9 wt.%, respectively.

An XPS test was conducted to study the element valence of S-3D-ZnO/rGO composites. The characteristic peaks of C, Zn, O and S are performed in the full spectrum (Figure 8a). The S 2p spectrum (Figure 8b) can be decomposed into four equivalent small peaks, at the binding energies of 162.1, 163.6, 164.8, and 168.8 eV, corresponding to the Zn-S, S 2p_3/2_, S 2p_1/2_, and sulfate bonds, respectively. In Figure 8c, the peaks at 1022.7 and 1045.7 eV correspond to Zn 2p_3/2_ and Zn 2p_1/2_, respectively. The spectrum of C 1s can be decomposed into five equivalent small peaks (Figure 8d). The equivalent substitution peaks at 284.6 and 284.8 eV correspond to the C=C bond and the C–C bond in the sp^2^ hybrid structure, respectively. The equivalent substitution peaks at 286.6 and 289.5 eV are attributed to C–O, C=O and O=C–O bonds, respectively, indicating that the obtained rGO after high temperature reduction still retains a certain number of oxygen-containing functional groups. The peak at 285.6 eV is due to the C–S bond, indicating a chemical bond between elemental sulfur and graphene. XPS results show that the sulfur in the S-3D-ZnO/rGO composite is mainly in the form of sulfur molecules, and a small amount of sulfur combines with ZnO or graphene to form specific chemical bonds. The formation of chemical bonds can enhance the combination of sulfur and ZnO/rGO composite, thus effectively improving the adsorption and immobilization of ZnO/rGO composite on active sulfur and polysulfide during the cycling process.

The study of the performance of the redox reaction of S-3D-ZnO/rGO composite was complemented with the CV test (Figure 9a). The CV curve shows two reduction peaks at about 2.06 and 2.32 V, corresponding to the reduction of S_8_ to long-chain polysulfides (Li_2_S_n_, 4 ≤ n ≤ 8) and the following reaction of the transformation from long-chain polysulfides to Li_2_S_2_/Li_2_S. During the subsequent reverse scanning, an oxidation peak at about 2.38 V is observed, which is due to the reverse reaction of the formation of S_8_ for the delithiation of Li_2_S and Li_2_S_2_. It is worth noting that the CV curves of the first three cycles perform a high coincidence, indicating the good reversibility and stability of the S-3D-ZnO/rGO electrode.

Constant current charge and discharge curves of the S-3D-ZnO/rGO and S-ZnO/rGO composites at 0.1 C are performed in Figure 9b,c. It can be seen that two discharge platforms appear at about 2.35 V and 2.10 V, and a long oxidation platform appears at around 2.40 V, which corresponds well with the CV result. The initial discharge capacities of S-ZnO/rGO and S-3D-ZnO/rGO cathodes are 1245 and 1277 mAh g^−1^, and discharge specific capacities in the third cycle are 1165 and 1204 mAh g^−1^. The capacity retention rates reached 93.5% and 94.3%, respectively. Compared with S-ZnO/rGO, S-3D-ZnO/rGO cathode shows higher initial discharge specific capacity and capacity retention for effective confinement on LiPS. The carefully introduced three-dimensional hollow structure perfectly encapsulates sulfur and prevents its diffusion into the electrolyte. At the same time, the polarization in S-3D-ZnO/rGO (ΔE = 0.21 V) is obviously suppressed compared with S-ZnO/rGO (ΔE = 0.28 V), which can be attributed to the enhanced effective utilization rate of ZnO in S-3D-ZnO/rGO. In the two-dimensional layered structure of ZnO/rGO, the graphenes are stacked on each other, such that the effective contact area of ZnO with polysulfide is reduced. The ZnO stacked inside the graphene sheet layer is not effectively utilized, and the chemical adsorption of ZnO cannot be not fully exerted. The three-dimensional hollow spherical structure of 3D-ZnO/rGO avoids the stacking of graphene, and the ZnO is fully exposed, providing sufficient active sites for the reaction of ZnO and LiPS, to produce effective suppressing on the shuttle effect.

Cycling performance of S-3D-ZnO/rGO and S-ZnO/rGO composite cathodes at 0.1 C is performed in Figure 9d. After 100 cycles, the discharge specific capacities of S-ZnO/rGO and S-3D-ZnO/rGO cathodes decreased to 838 and 949 mAh g^−1^, and the corresponding capacity retention rates are 67.3% and 74.3%, respectively. Furthermore, the as-developed S-3D-ZnO/rGO cathode can fulfill decent sulfur electrochemistry, even under a high sulfur loading of 4.5 mg cm^−2^ (Appendix A). It can be seen that the cycle performance of S-3D-ZnO/rGO is more stable. The higher stability can be attributed to the following reasons: (1) the 3D-ZnO/rGO can coat sulfur inside the hollow spherical structure, thereby reducing the dissolution of LiPS into the electrolyte [33,34,35,36]. (2) The internal space of the hollow spherical structure can effectively alleviate the volume change of sulfur during the cycling process, resulting in the high structural integrity of the electrode. We can intuitively observe that 3D-ZnO/rGO has stronger adsorption capacity for polysulfide through the adsorption experiment in Appendix A; secondly, we can also observe the volume change of electrode material after the cycling experiment through SEM after cycling. As shown in Appendix A, at 1 C after 100 cycles, the morphology of 3D-ZnO/rGO is still a hollow sphere, while the morphology of ZnO/rGO has collapsed.

The same advantage of the S-3D-ZnO/rGO cathode is demonstrated in the rate performance test (Figure 9e). As the current density gradually increases, the discharge specific capacities of both gradually decrease, which is caused by high polarization, low sulfur utilization and limited charge transfer at high current density. At 1 C, the discharge specific capacities of S-ZnO/rGO and S-3D-ZnO/rGO are 654 and 726 mAh g^−1^, respectively. When the current density is returned from 1 C to 0.1 C, the discharge capacities of 987 and 1034 mAh g^−1^ are obtained again. Compared with the S-ZnO/rGO cathode, S-3D-ZnO/rGO exhibits a higher discharge specific capacity under the same test conditions. This is because the spherical structure of 3D-ZnO/rGO can maintain structural integrity even at a high c-rate, resulting in effective suppression of the shuttle effect.

In order to further understand the electrochemical performance of the S-3D-ZnO/rGO composite, the EIS results of S-3D-ZnO/rGO cathode before cycling and at the 10th cycle were tested in the frequency range 100 kHz to 0.01 Hz (Figure 9f). After ten cycles, the charge transfer resistance of the battery was reduced from 27 Ω to 15 Ω. The high resistance before cycling for the cathode material is perhaps not sufficiently contacted with the electrolyte. However, as the cycle progresses, the cathode material is more in contact with the electrolyte, thereby lowering the charge transfer resistance of the battery. In addition, after 10 cycles, the slope of the EIS in the low frequency region increased, indicating that the Warburg impedance in the battery is reduced.

The working mechanism between the ZnO and LiPS was explored by density functional theory (DFT) calculations. The optimized geometrical configuration of Li_2_S_6_-ZnO is shown in Figure 10a. The configuration demonstrated the bond of the S atoms in Li_2_S_6_ and the Zn atoms on ZnO (101) surface, which was consistent with the above XPS result. At the same time, a high binding energy of −2.49 eV was achieved, indicating the strong adsorption of ZnO toward LiPS. Apart from the strong adsorption effect, ZnO also demonstrated the ability to boost the sulfur species decomposition. As shown in Figure 10b, a small energy barrier of 0.12 eV for Li_2_S decomposition was performed and the corresponding geometrical configurations were displayed in Figure 10c, which confirmed the remarkable catalysis ability of ZnO on sulfur species transformation.

## 4. Conclusions

In summary, an effective sulfur host (3D-ZnO/rGO) is prepared by a simple method. The obtained 3D-ZnO/rGO possesses spherical hollow structure, which is beneficial for the sulfur loading. ZnO can effectively confine LiPS by Zn-S bond and prevent the dissolution of LiPS into the electrolyte. At the same time, the 3D-rGO can immobilize LiPS by physical adsorption. The obvious relief on the volume change can also be realized by the designed structure. When used as a cathode, the polarization is relieved in S-3D-ZnO/rGO electrode, and the excellent cycling stability and outstanding rate performance are achieved. The results show that S-3D-ZnO/rGO is an ideal cathode material for LSB.

## Figures and Tables

**Figure 1 nanomaterials-10-01633-f001:**
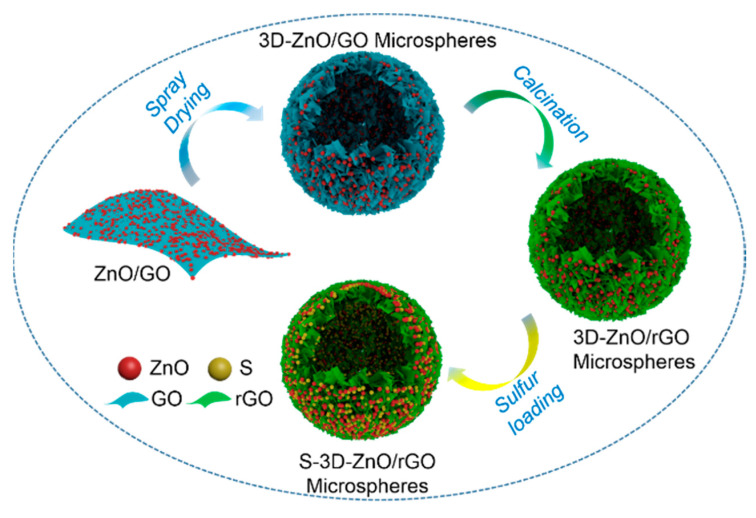
Schematic illustration for the synthesis of S-3D-ZnO/rGO composite.

**Figure 2 nanomaterials-10-01633-f002:**
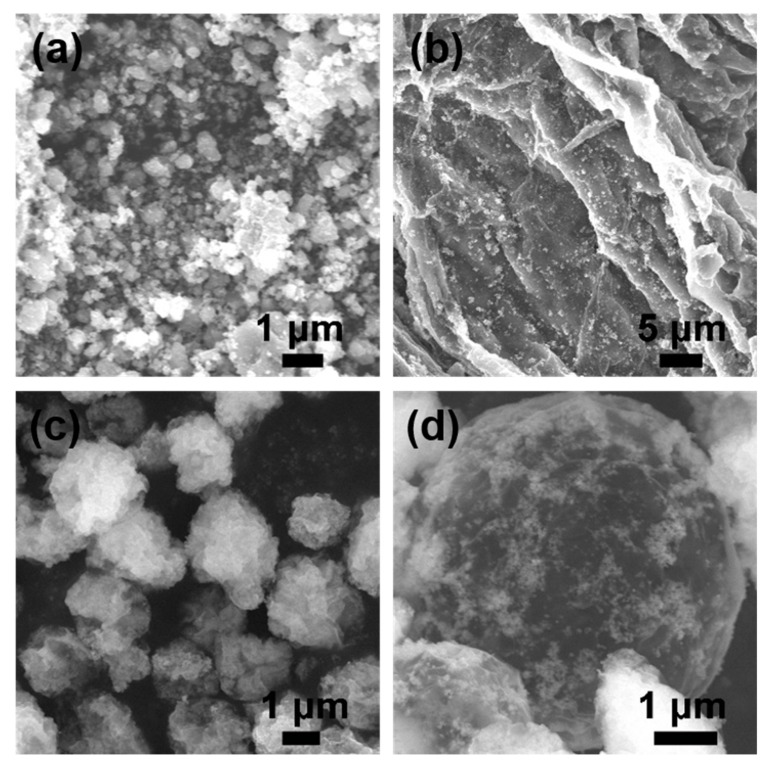
SEM images of (**a**) pure ZnO; (**b**) ZnO/rGO; (**c**,**d**) 3D-ZnO/rGO.

**Figure 3 nanomaterials-10-01633-f003:**
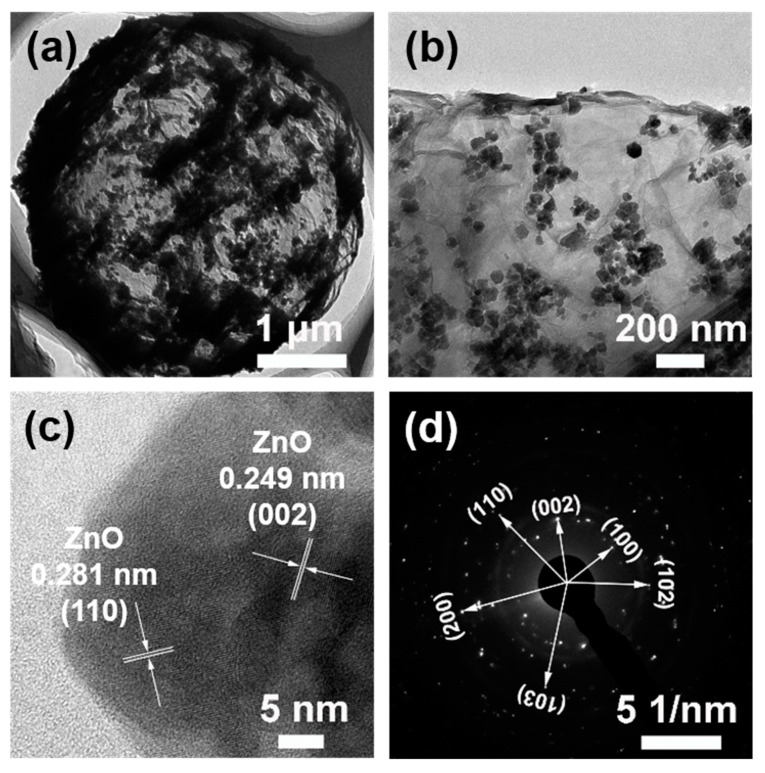
(**a**,**b**) TEM images of 3D-ZnO/rGO. (**c**) high resolution TEM image of 3D-ZnO/rGO. (**d**) the selected area electron diffraction (SAED) pattern of 3D-ZnO/rGO.

**Figure 4 nanomaterials-10-01633-f004:**
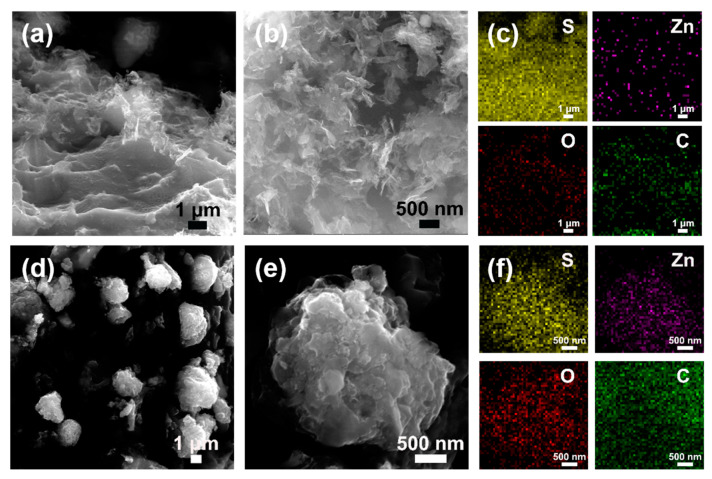
SEM images of S-ZnO/rGO (**a**,**b**) and corresponding element distribution (**c**). SEM images of S-3D-ZnO/rGO (**d**,**e**) and corresponding element distribution (**f**).

**Figure 5 nanomaterials-10-01633-f005:**
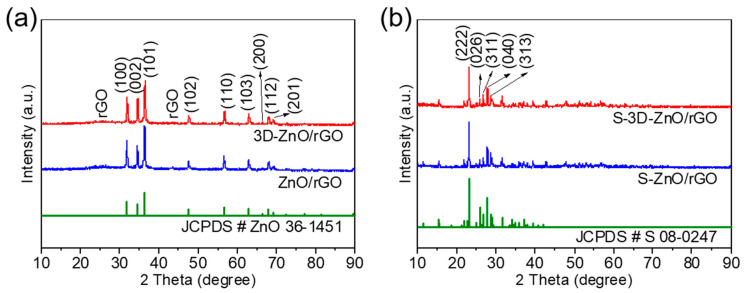
XRD spectra of (**a**) ZnO/rGO and 3D-ZnO/rGO composites, (**b**) S-ZnO/rGO and S-3D-ZnO/rGO composites.

**Figure 6 nanomaterials-10-01633-f006:**
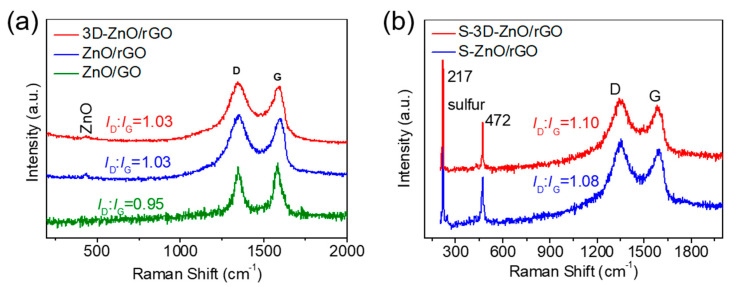
Raman spectra of (**a**) ZnO/GO, ZnO/rGO and 3D-ZnO/rGO composites, (**b**) S-ZnO/rGO and S-3D-ZnO/rGO composites.

**Figure 7 nanomaterials-10-01633-f007:**
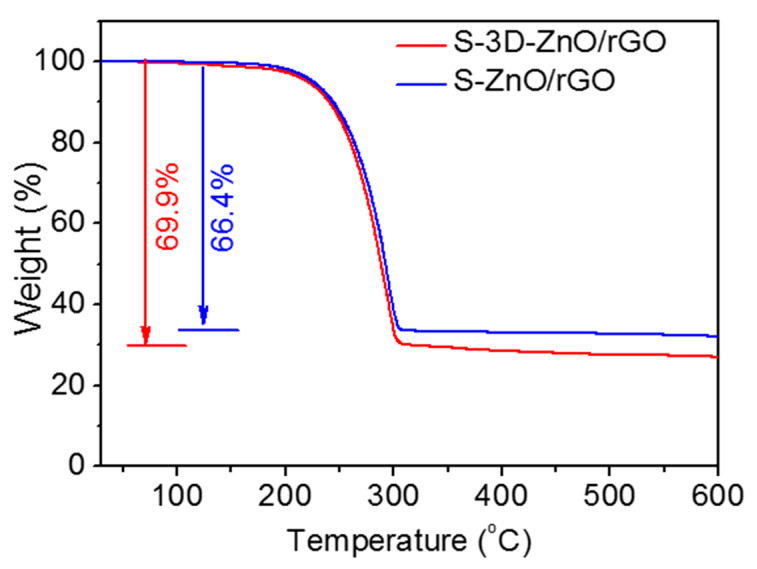
TGA curves of S-ZnO/rGO and S-3D-ZnO/rGO composites.

**Figure 8 nanomaterials-10-01633-f008:**
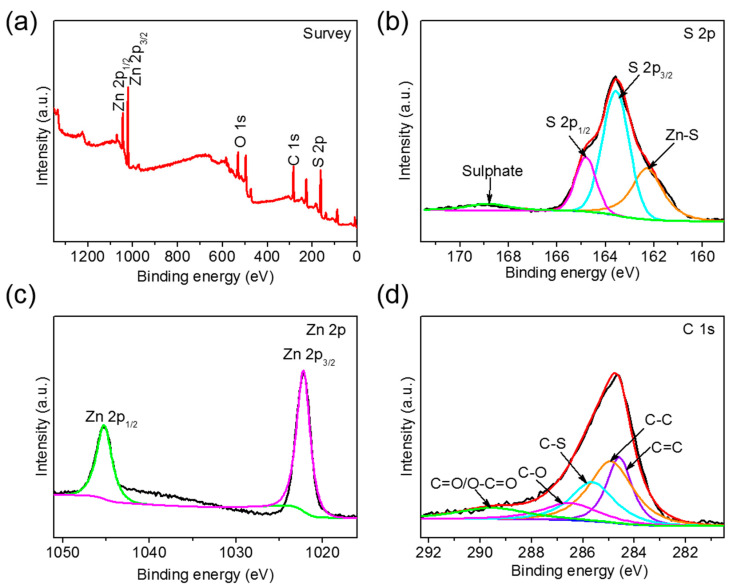
(**a**) XPS full spectrum, (**b**) S 2p (The orange peak at 162.1 eV, blue peak at 163.6 eV, purple peak at 164.8 eV, and green broad peak at 168.8 eV correspond to Zn-S, S 2p3/2, S 2p1/2 and sulfate bonds, respectively.), (**c**) Zn 2p (The purple peak at 1022.7 eV and the green peak at 1045.7 eV correspond to Zn 2p3/2 and Zn 2p1/2.) and (**d**) C 1s of S-3D-ZnO/rGO composite.

**Figure 9 nanomaterials-10-01633-f009:**
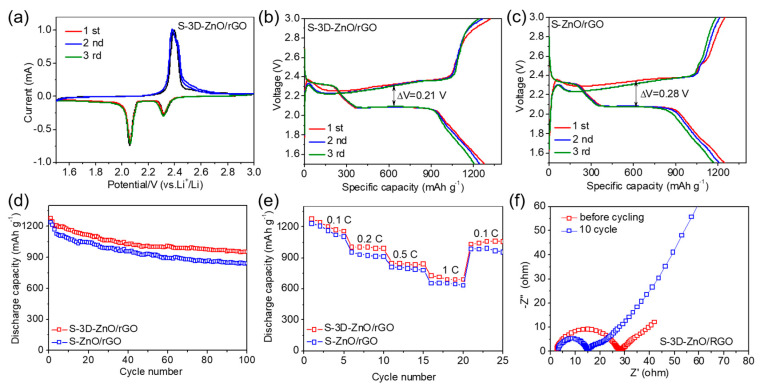
(**a**) CV curves of S-3D-ZnO/rGO composite cathode. Constant current charge and discharge curves of (**b**) S-3D-ZnO/rGO and (**c**) S-ZnO/rGO composites at 0.1 C. (**d**) Cycling performance of S-3D-ZnO/rGO and S-ZnO/rGO composite cathodes at 0.1 C. (**e**) Rate performance of S-3D-ZnO/rGO and S-ZnO/rGO composite cathodes. (**f**) EIS spectra of S-3D-ZnO/rGO composite cathode before cycling and at 10th cycle.

**Figure 10 nanomaterials-10-01633-f010:**
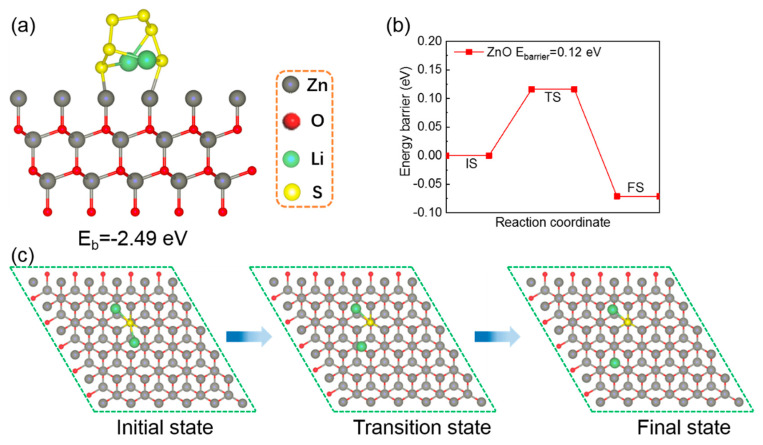
(**a**) The optimized geometrical configuration of Li_2_S_6_-ZnO; (**b**) energy profiles of Li_2_S decomposition on ZnO (101) surface; (**c**) the geometrical configurations of Li_2_S decomposition on ZnO (101) surface.

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
