# Peer review of "3D Hollow rGO Microsphere Decorated with ZnO Nanoparticles as Efficient Sulfur Host for High-Performance Li-S Battery"

_nanomaterials, 2020, doi:10.3390/nano10091633_

Round 1

Reviewer 1 Report

Title of manuscript: 3D Hollow rGO Microsphere Decorated with ZnO Nanoparticles as Efficient Sulfur Host for High-performance Li-S battery.

In this manuscript, authors have reported a comprehensive research on the performance of Li-S battery by developing 3D hollow rGO microsphere decorated with ZnO nanoparticles as efficient sulfur host. The objective of this manuscript is clear and provided some meaningful data to clarify their objective. The manuscript is well organized. I think authors need to do the English correction and provide some supporting data as well.

Comments:

  1. Authors are requested to revise the spelling, grammatical errors, format mismatching throughout the manuscript. Particularly, mentioned bellow:
  2. Revise the word “Porphyrinderived” (LN 44, PN-2).
  3. In the manuscript there are somewhere “Capacity” and “Specific capacity” also “Cycle stability” and “Cycling stability”. It will be better to write only “Specific capacity” and “Cycling stability” throughout the manuscript.
  4. The sentence “After sulfur loading, the obtained S-3D-ZnO/rGO is used as cathode” is not correct. It should be changed to “After sulfur loading, the obtained S-3D-ZnO/rGO is used as a/the cathode”
  5. In figure 2, 4 the color of figure number (a, b, c, and d) is different. It could be better to put the same color and clearly visible number.
  6. In figure 3(c) the high resolution TEM image of 3D-ZnO/rGO, the lattice spacing is not clearly visible, it will be better to change this image with more clear high resolution TEM image.
  7. In figure 5(a) and 5(b), authors should include the reference JCPDS data with sample data in same graph rather mentioning reference characteristics peaks in literature.
  8. In figure 9 (b) and (c) charge discharge curve has capacity on the X axis. It will be better to change it to “specific capacity”. Also in figure 9(f) and literature (LN 279 and 281, PN 10) authors used “0 cycle”. It could be better to change it to “before cycling”.
  9. According to the author, the pure ZnO has low specific surface area and low adsorption capacity for sulfur and LiPS than 3D hollow spherical structure. How author can explain this without any data regarding surface area analysis of them?
  10. According to the author after the third cycle, the capacity retention increased due to the prevention of diffusion of the sulfur into electrolyte by three-dimensional hollow structure. But there is no supporting data to clarify this phenomenon. It will be better to provide some relevant data (XRD, FTIR, or other appropriate analysis) after third cycle study to see that there are sulfurs in the electrolyte or not.
  11. In figure 9(d), the author claimed that the cycling performance of S-3D-ZnO/rGO is more stable than S-ZnO/rGO due to the reducing dissolution of LiPS into the electrolyte and alleviate the volume change of sulfur during the cycling process. How author can conclude this without providing any relevant data regarding polysulfide dissolution and the volume change of sulfur after cycling test?
  12. In conclusion the author has concluded that, “ZnO can effectively confine LiPS by O-S bond and prevent the dissolution of LiPS into the electrolyte”. Author needs to provide the XPS or other appropriate data to support this statement.

Reviewer 2 Report

In this contribution, rGO microspheres decorated with ZnO particles are prepared by spray-drying and then composited with S by melt diffusion method. These materials are characterized by different method: SEM, TEM, XRD, XPS, Raman, TG and cyclic voltammetry to evaluate their potential application as cathode in lithium-sulphur batteries. The following issues should be considered before publication.

  1. It is recommendable to analyze the SEM and TEM image by EDX method to better visualize the elemental distribution. In the actual image, it is difficult to discern the ZnO particles.
  2. The phase composition by XRD needs to be improved. After sulphur loading new diffraction peaks are observed, which need to be identified.
  3. The bands assigned to S-S should be indicated in the Fig. 6b.
  4. The sulphur content estimated by TGA is very high up to 70 wt.%. It is not clear if sulphur is oxidized into SO2 at 200 ºC in N2 atmosphere. This should be confirmed by XRD analysis, the original pattern of the Fig. 5a should be recovered.

The authors deals that he Warburg impedance is reduced after 10 cycles. By assuming that the processes in Fig. 9f are the same, it is clear that the polarization resistance increases after 10 cycles. It is also recommendable to include frequency values in the spectra.

Reviewer 3 Report

The article entitled 3D Hollow rGO Microsphere Decorated with ZnO Nanoparticles as Efficient Sulfur Host for High- performance Li-S battery needs major revision before to be considered for publication Nanomaterials MDPI journal

Here my comments:

1) A discussion about polysulfides retention is necessary and some articles must be cited: a) J. Mater. Chem. A, 7, 2019, 12381-12413 b) Journal of Power Sources 319, 2016, 247-254; c) Journal of Power Sources 344, 2017, 208-217; d) Sustainable Energy & Fuels 1 (4),2017, 737-747. The authors must describe a possible mechanism differences about 3D ZnO-rGo vs ZnO-rGO

2) Some articles about Li-S must be cited:a) Adv. Funct. Mater. 2020, 2001812; b) J. Am. Chem. Soc. 2020, 142, 7, 3583–3592; c) The Journal of Physical Chemistry C 122 (2), 2018, 1014-1023; d) Nano Energy 76 (2020) 105033, e) Journal of Power Sources 427, 2019, 201-206; f) Nanoscale, 2020,12, 13980-13986

3) About the material proposed: is it possible to prepare other metal oxide-rGO microspheres?

4) What is the final % of sulfur as active material in the cathode? Is it possible to increase the sulfur loading over 3 mg/cm2?

Round 2

Reviewer 1 Report

The authors did good work to answer my comments, which are all acceptable. The manuscript is now good enough to be accepted for publication in Nanomaterials. 

Reviewer 2 Report

The authors have improved the manuscript and the paper can be accepted.

Reviewer 3 Report

I recommend to accept the paper in the present form